# Structural Characteristics and Emulsifying Properties of Soy Protein Isolate Glycated with Galacto-Oligosaccharides under High-Pressure Homogenization

**DOI:** 10.3390/foods11213505

**Published:** 2022-11-03

**Authors:** Yitong Xie, Rongxu Liu, Changge Zhang, Danyi Liu, Jianchun Han

**Affiliations:** 1College of Food Science, Northeast Agricultural University, Harbin 150030, China; 2Heilongjiang Green Food Science Research Institute, Harbin 150030, China

**Keywords:** high-pressure homogenization, soy protein isolate, glycation, galacto-oligosaccharides, emulsifying properties

## Abstract

This study explored the Maillard reaction process during the glycation of soy protein isolate (SPI) with galacto-oligosaccharides (GOSs) under high-pressure homogenization (HPH) and its effects on the emulsifying properties of SPI. SPI-GOS glycation under moderate pressure (80 MPa) significantly inhibited the occurrence and extent of the Maillard reaction (*p* < 0.05), but homogenization pressures in the range of 80–140 MPa gradually promoted this reaction. HPH caused a decrease in the surface hydrophobicity of the glycated protein, an increase in the abundance of free sulfhydryl groups, unfolding of the protein molecular structure, and the formation of new covalent bonds (C=O, C=N). Additionally, the particle size of emulsions created with SPI-GOS conjugates was reduced under HPH, thus improving the emulsifying properties of SPI. A reduction in particle size (117 nm), enhanced zeta potential (−23 mV), and uniform droplet size were observed for the emulsion created with the SPI-GOS conjugate prepared at 120 MPa. The conformational changes in the glycated protein supported the improved emulsification function. All results were significantly different (*p* < 0.05). The study findings indicate that HPH provides a potential method for controlling glycation and improving the emulsifying properties of SPI.

## 1. Introduction

Soy protein isolate (SPI) is a commercially important biopolymer with a protein concentration ≥ 90% on a dry weight basis [1]. SPI is commonly employed in food compositions because of its excellent nutritional value, functionality, and health benefits [2]. Indeed, properties such as film formation, emulsification, gelation, and water binding favor the application of SPI in the food industry [3]. Considering its large molecular mass and considerable number of hidden hydrophobic groups, SPI is considered to be an efficient food emulsifier if suitably modified and a good choice for delivering bioactive chemicals [4,5].

Glycation is an effective means of improving the functional properties of proteins, including their emulsification function, protein solubility, and thermal stability [6,7]. However, the Maillard reaction at the end of the glycation process typically produces harmful products such as advanced glycation end-products (AGE) and colored nitrogenous polymers and co-polymers [8]. Therefore, controlling the degree of response during the early stages of conjugation is important. The effects of reducing sugar type and treatment method on the degree of the Maillard reaction have been a main focus of research [9,10].

Galacto-oligosaccharides (GOSs) are oligosaccharides composed of galactose and glucose groups. When glycated with proteins, GOSs can effectively reduce the allergenicity of proteins and improve their functional properties. Seo et al. [11] reported that lysozymes (LZMs) glycated with GOS had greater emulsion stability than LZM-galactan and LZM-galactose conjugates. Further, Peled and Livney [12] demonstrated that GOS-lactoferrin hydrolysate conjugates self-assembled into 10–25 nm particles that exhibited good colloidal stability and were less sensitive to proteolysis than native lactoferrin.

High-pressure homogenization (HPH) is a simple procedure using mild processing conditions and a short treatment period [13]. HPH is based on the continuous flow of a liquid through a narrow hole, which increases velocity while simultaneously creating a high-pressure drop to atmospheric pressure, resulting in high turbulence, shear stress, temperature rise, and cavitation effects [14]. Numerous research groups have studied the effects of HPH on SPI structure and function. For example, protein unfolding was favored up to a homogenization pressure of 100 MPa, which exposed the hydrophobic groups; however, unfolded proteins reassembled through interparticle interactions beyond this pressure threshold [15]. HPH has been shown to modify lentil protein isolate through functional, structural, and rheological changes that provide better solubility, as well as improved emulsifying, foaming, and gelling capabilities [16]. Shi et al. [17] reported that HPH treatment decreased the size of dispersed phase particles and increased emulsion stability. Although several studies have investigated HPH treatment of various protein isolates, the use of HPH to elicit a Maillard reaction and hence modify the structure and emulsifying function of SPI has not been reported.

We speculated that SPI and GOS under the action of HPH may cause the Maillard reaction to occur during glycation, resulting in changes in protein structure and emulsifying properties. Therefore, the goal of the study was to explore the effect of HPH on the Maillard reaction process, hoping to slow the occurrence of the reaction, control it in early and middle stages, and inhibit the production of harmful substances. Additionally, we aimed to compare the effects of HPH and wet heating on SPI glycation.

## 2. Materials and Methods

### 2.1. Materials

Low temperature-defatted soybean meal was obtained from our laboratory (Harbin, China). GOSs were obtained from Shanghai Yuanye Bio-Technology Co., Ltd. (Shanghai, China). The SDS-PAGE kit containing 30% Acr-Bis (29:1), 1.5 M Tris-HCl (pH 8.8), 1.0 M Tris-HCl (pH 6.8), and 10% SDS was obtained from Solarbio Life Sciences (Beijing, China). The remaining reagents were of analytic grade.

### 2.2. Preparation of SPI

SPI was prepared according to the method described by [18], with some modifications. Soybean meal was dissolved in deionized water for 1 h (1:10, *w*/*v*), adjusted to pH 8.0 with 2 M NaOH, stirred for 1 h, and centrifuged (4 °C, 9500× *g*, 20 min). The supernatant was adjusted to pH 4.5 with 2 M HCl and centrifuged (4 °C, 9500× *g*, 20 min). The protein precipitate was washed with deionized water and neutralized to pH 7.0 with 2 M NaOH. Subsequently, the SPI was freeze-dried and stored at -20 °C until further use. The protein content of SPI was determined to be 91.75% (*w*/*w*) using the biuret method [18].

### 2.3. Preparation of SPI-GOS Conjugates by HPH

SPI solution (2% *w*/*v*) and GOS (3:1 *w*/*w*) were dissolved in 0.05 mol/L phosphate buffer solution (pH 8), stirred at room temperature for 2 h, and stored at 4 °C for 24 h to ensure full hydration. Samples underwent traditional wet heating to induce the Maillard reaction, HPH treatment at various homogenization pressures, and no treatment (control groups) as follows:(1)Traditional wet heating Maillard reaction group (0.1 MPa): SPI-GOS mixed solution was heated in a water bath at 70 °C for 10 min to obtain SPI-GOS conjugates. This sample was named SPI-GOS-0.1.(2)HPH treatment groups: SPI-GOS mixed solution was subjected to HPH (ATS Engineering Limited, Suzhou, China) at 70 °C under 80, 100, 120, and 140 MPa for 10 min to obtain SPI-GOS conjugates. The samples were named SPI-GOS-80, SPI-GOS-100, SPI-GOS-120, and SPI-GOS-140.(3)Control groups: native 2% (*w*/*v*) SPI solution (SPI) and a mixture of SPI and GOS (SPI-GOS-M) were employed as controls.

All samples were promptly chilled in an ice-water bath and freeze-dried.

### 2.4. Measurement of Intermediate Products and Degree of Browning

The measurement of Maillard reaction products (MRPs) was performed as described by [19], with slight modifications. Samples were diluted with 0.1% SDS to a protein concentration of 5 mg/mL, and the absorbances of the intermediate and final MRPs were measured using a UV-visible spectrophotometer (Shimadzu UV-2600; Shimadzu Corp., Kyoto, Japan) at 294 and 420 nm, respectively.

### 2.5. Measurement of Free Amino Acids

The free amino acid content was determined using the ortho-phthaldialdehyde (OPA) method [20]. Samples were diluted with deionized water to a protein concentration of 2 mg/mL, and 200 μL diluted sample was mixed with 4 mL OPA at 40 °C. After 2 min, the absorbance of the sample was measured at 340 nm using a UV-visible spectrophotometer (Shimadzu UV-2600; Shimadzu Corp.). A standard curve was constructed using L-lysine.

### 2.6. SDS-PAGE

SDS-PAGE was performed according to the method described by [18], with some modifications. The samples were diluted with sample buffer to a protein concentration of 2 mg/mL and heated in a water bath at 90 °C for 5 min. The stacking and separation gels contained 5 and 12% acrylamide, respectively, and the sample volume was 10 μL. Electrophoresis was performed at 80 V for 15 min, and then adjusted to 120 V for 30 min. After electrophoresis, the gel was stained for 10 min and decolorized.

### 2.7. Measurement of Surface Hydrophobicity

Surface hydrophobicity was measured using 1-aniline-8-naphthalenesulfonate (ANS) as a fluorescent probe, as described by [21], with some modifications. The sample was diluted to a protein concentration of 0.025–0.2 mg/mL with 0.01 mol/L phosphate buffer (pH 7.0). A 4 mL aliquot of sample was mixed with 50 μL ANS. The fluorescence intensity was measured using a fluorescence spectrophotometer at excitation and emission wavelengths of 390 and 470 nm, respectively. The fluorescence intensity was plotted against the protein concentration, and the surface hydrophobicity index was calculated as the slope of the curve.

### 2.8. Measurement of Sulfhydryl Content

Measurement of the sulfhydryl content was conducted as described by [22,23]. Ellman’s reagent was used to calculate the total sulfhydryl (T-SH) and surface free sulfhydryl (S-SH) contents. Determination of T-SH was performed as follows: 2 mL sample (10 mg/mL) was added to 10 mL Tris-glycine buffer (0.086 mol/L Tris, 0.09 mol/L glycine, 8 mol/L urea, 0.04 mol/L EDTA) and 0.8 mL DNTB (4 mg/mL). After mixing and standing for 15 min, the absorbance of the sample was measured at 412 nm. Determination of S-SH was performed as follows: 2 mL sample (10 mg/mL) was added to 10 mL Tris-glycine buffer (0.086 mol/L Tris, 0.09 mol/L glycine, 0.04 mol/L EDTA) and 0.8 mL DNTB (4 mg/mL). After mixing and standing for 15 min, the absorbance of the sample was measured at 412 nm. The SH content was calculated using Equation (1):(1)$$\mathrm{SH}\;(\mu \mathrm{mol}/g)\;=\frac{A_{412}\times D}{\varepsilon \times C}$$
where *A*_412_ is the absorbance value at 412 nm, *C* is the sample concentration (mg/mL), *ԑ* is a molar extinction coefficient of 13,600 mol^−1^/cm, and *D* is the dilution factor.

### 2.9. Circular Dichroism (CD) Spectroscopy

The secondary structure of the protein was assessed using circular dichroism (CD) spectroscopy according to the method described by [24], with some modifications. The CD measurements were performed using the J-815 spectropolarimeter (Jasco Corp., Tokyo, Japan) using a quartz cell with a path length of 1.0 mm and a temperature of 25 °C to scan the protein solutions (1% *w*/*v*) from 190 to 260 nm. The scanning speed was 100 nm/min with a step size of 1 nm and a bandwidth of 1.0 nm. The protein secondary structure was estimated from the obtained spectra using the CDPro software package.

### 2.10. Fourier Transform Infrared (FT-IR) Spectroscopy

The structure of the protein was further analyzed using a Nicolet iS5 FT-IR spectrometer (Thermo Fisher Scientific, Waltham, MA, USA). The freeze-dried samples were combined with potassium bromide to obtain a powder, which was then crushed into a tablet. Spectrum acquisition was performed at 2 cm^−1^ resolution in the 4000–500 cm^−1^ range, using potassium bromide as a blank control.

### 2.11. Intrinsic Fluorescence Spectroscopy

The tertiary structure of the protein was assessed using endogenous fluorescence spectroscopy according to the method described by [25], with slight modifications. The sample was diluted with 0.01 M phosphate buffer (pH 7.0) to a protein concentration of 1 mg/mL. Samples were stimulated at 295 nm (slit = 5 nm) and emission spectra were collected at 1200 nm/min from 310 to 450 nm (slit = 5 nm).

### 2.12. Preparation of Emulsions

Freeze-dried SPI, SPI-GOS-M, and SPI-GOS conjugate powders were dissolved in 0.01 mol/L phosphate buffer (pH 7.0) to a protein concentration of 10 mg/mL. Soy oil was added at a protein:oil volume ratio of 3:1 and then homogenized using a high-speed homogenizer (10,000 rpm, 1 min) at room temperature.

### 2.13. Measurement of Emulsion Activity and Stability

The emulsifying activity index (*EAI*) and emulsion stability index (*ESI*) were calculated using the approach described by [26], with slight adjustments. A 100 μL aliquot of emulsion was diluted 100 times with 0.1% SDS solution and fully mixed. The absorbance of SDS solution (*A*_0_) was determined at 500 nm as the blank. The absorbance *A*_10_ of the sample (*A*_10_) was obtained after 10 min at 500 nm. *EAI* was calculated using Equation (2):(2)$$EAI\left(m^{2}/g\right)=2T\frac{A_{0}\times N}{C\times U\times 1000}$$
where *T* is constant with a value of 2.303; *N* is the dilution ratio of 100; *C* is the protein concentration before emulsion formation (g/mL); and *U* is the oil volume fraction 0.25. *ESI* was calculated using Equation (3):(3)$$ESI\left(\min\right)=\frac{A_{0}}{A_{0}-A_{10}}\times 10$$

### 2.14. Measurement of Particle Size

Emulsion samples were diluted with 0.01 mol/L phosphate buffer (pH 7.0) to a protein concentration of 1 mg/mL. A Zetasizer Nano instrument (Malvern Instrument Co., Ltd., Worcestershire, UK) was used to determine the average particle size, particle size distribution, and polydispersity index (PDI) of the emulsions.

### 2.15. Measurement of Zeta Potential

Emulsion samples were diluted with 0.01 mol/L phosphate buffer (pH 7.0) to a protein concentration of 1 mg/mL. The emulsions were injected into a capillary absorption kettle, and the zeta potential was monitored using a Zetasizer Nano instrument (Malvern Instrument Co.). Measurements were performed in triplicate.

### 2.16. Confocal Laser Scanning Microscopy (CLSM)

The microstructure of the emulsion was assessed using a Leica TCS SP2 microscope (Leica Microsystems Inc., Heidelberg, Germany). Nile red staining solution (50 μL) and fluorescence amine staining solution (40 μL) were added to 2 mL emulsion, the sample was mixed, and then stored away from light for 30 min. A 4 μL aliquot of the treated emulsion was dropped onto the pre-cleaned slide, spread evenly, gently covered with a coverslip to avoid air bubbles, and the edges were sealed with nail polish. Finally, a confocal image of the emulsion was obtained using a 40× objective lens. The excitation wavelengths of the oil droplet and protein were 488 and 543 nm, respectively.

### 2.17. Statistical Analysis

The experimental data were analyzed and plotted using IBM SPSS Statistics v.20 software (IBM Corp., Armonk, NY, USA) and Origin v. 9.0 software (OriginLab, Corp., Northampton, MA, USA). The results of three replicate independent tests were expressed as the mean ± standard deviation. A *p*-value < 0.05 was considered to be statistically significant.

## 3. Results

### 3.1. Effect of HPH on Conjugation of SPI and GOS

#### 3.1.1. Intermediate MRPs and Browning of SPI-GOS Conjugates

During the initial stage of the Maillard reaction, the products of sugar-amine condensation and Amadori rearrangement are colorless. These products have a maximum absorption in the ultraviolet region (approximately 294 nm) [27]. As the Maillard reaction enters the final stage, aldol condensation, aldehyde-amine condensation, and the formation of heterocyclic nitrogen compounds lead to browning [27], the degree of which is reflected by increased absorbance at 420 nm. As shown in Figure 1, the quantity of intermediate MRPs increased and then decreased with increasing homogenization pressure. The absorption value of the SPI-GOS conjugates reached a maximum when treated at 0.1 MPa and decreased significantly when treated at 80 MPa (*p* < 0.05), indicating that HPH slowly promoted the occurrence of the Maillard reaction. Huang et al. [10] reported that the production of intermediate MPRs decreased when the dynamic high-pressure microfluidization pressure exceeded 100 MPa. However, the intermediate MRP content and degree of browning increased at homogenization pressures of 80–140 MPa. Higher pressure causes the molecular structure of the protein to unfold, thus increasing its reactivity. HPH reportedly caused further unfolding of lentil protein isolate to increase protein reactivity [14]. These results indicated that increasing homogenization pressure gradually promoted the Maillard reaction. In addition, the intermediate MRP content decreased while the degree of browning increased when the pressure increased from 120 to 140 MPa, indicating that high homogenization pressure accelerated the transition of the Maillard reaction from initial to final stages. In summary, the results indicate that HPH can slowly promote the Maillard reaction.

#### 3.1.2. Free Amino Acid Content

The Maillard reaction between the carbonyl group of the reducing sugar and the amino group of the protein leads to protein glycation [28]. Arginine and lysine play important roles in the glycation process [20]. As shown in Figure 2, compared to SPI and SPI-GOS-M, the SPI-GOS conjugates demonstrated a decrease in free amino acid content. Due to the consumption of free amino acids, the SPI-GOS conjugates displayed greater reactivity during the glycation process. These results concurred with those reported by [20,29].

Additionally, the free amino acid content of the HPH-treated SPI-GOS conjugates decreased with increasing homogenization pressure. Stress appears to have a favorable effect on the Maillard process. Stress-induced conformational changes in SPI, where the structure becomes loose and the number of glycation sites increases, is known to lead to a decrease in the free amino acid content. The reaggregation of unfolded proteins under high pressure, once a specified pressure is attained, could explain the gradual flattening of the decreasing free amino acid content curve [30]. Additionally, the free amino acid content of SPI-GOS-0.1 was significantly lower than that of the HPH-treated SPI-GOS conjugates (*p* < 0.05). This outcome aligned with the intermediate MRP and degree of browning results.

#### 3.1.3. SDS-PAGE Results

The formation of MRPs was demonstrated using SDS-PAGE [31]. The electrophoretic profiles of the SPI−GOS conjugates and MRPs under various treatment conditions are shown in Figure 3. Five protein bands were observed for the original SPI (lane 1), corresponding to subunits of the 7S fraction, including the *α*′, *α* (66.2 kDa), and *β* (44.3 kDa) subunits, and the 11S fraction, including the A (29–44.3 kDa) and B (20.1 kDa) subunits [32]. When the pressure was 0.1–140 MPa, a new band at the upper end of the gel was observed, and the band deepened with the increase in pressure. It indicated that SPI and GOS underwent the Maillard reaction to create the large molecular weights of the conjugates, which are too large to migrate into the separation gel. In addition, the disappearance of the 60 kDa and 19 kDa bands indicated that the subunits of the corresponding molecular weight proteins covalently combined with the sugar molecules to form the conjugate. Similarly, Ma et al. [33] reported that ultrasound−assisted Maillard reaction resulted in the appearance of new protein bands for SPI and disappearance of some existing bands.

### 3.2. Effect of HPH on Changes in SPI-GOS Conjugate Physicochemical Properties

#### 3.2.1. Surface Hydrophobicity

Hydrophobic forces are primarily expressed by surface hydrophobicity (H_0_), which is one of the main physical features of proteins that impacts their emulsifying properties [34]. The surface hydrophobicity of a protein is altered as a result of conformational and functional changes, which require the exposure of hydrophobic amino acids on the protein surface [35]. Surface hydrophobicity represents the degree of intramolecular hydrophobic group exposure, implying that higher surface hydrophobicity is achieved when more hydrophobic groups are exposed [36].

The H_0_ values of SPI and the SPI-GOS conjugates are shown in Figure 4. The H_0_ values of the SPI-GOS conjugates were significantly lower than that of SPI (*p* < 0.05). The H_0_ of SPI-GOS-0.1 was significantly lower than that of the SPI-GOS conjugates treated with HPH treatment (*p* < 0.05). This was consistent with the extent of the Maillard reaction. In addition, the H_0_ values of the HPH-treated SPI-GOS conjugates decreased with increasing homogenization pressure, achieving the lowest H_0_ at 140 MPa. The protein molecules were denatured and unfolded by HPH treatment, and a large number of hydrophobic groups were rearranged to form internal hydrophobic regions as the Maillard reaction progressed [37]. Moreover, the SPI molecules unfolded and accessed the GOS-containing polyhydroxy groups, which increased the number of hydrophilic groups and reduced the surface hydrophobicity of the MRPs, leading to a decrease in H_0_ values and greatly improving the hydrophilicity of the proteins [13]. In summary, the results indicate that HPH combined with the Maillard reaction can effectively reduce the surface hydrophobicity of SPI, improve the protein’s hydrophilicity, and further improve emulsification.

#### 3.2.2. Sulfhydryl Content

The structure of a protein is directly related to its emulsifying properties, and the sulfhydryl content is an important indicator of protein structure [38]. Figure 5 and Figure 6 show the T-SH and S-SH contents of the SPI-GOS conjugates. As shown in Figure 5, the S-SH content of the SPI-GOS conjugates was higher than that of SPI and SPI-GOS-M. The highest S-SH concentration (12.12 ± 0.50 μmol/g) was achieved at 0.1 MPa. The increased S-SH content of the SPI-GOS conjugates could be attributed to surface SH-group exposure caused by aggregation, fragmentation, or glycation following HPH treatment [22]. Shen and Tang [39] reported that the abundance of free SH groups decreased in SPI treated at 120 MPa, which was attributed to the production of new disulfide bonds via SH/SS intramolecular or intermolecular interactions. As shown in Figure 6, the T-SH content decreased with increasing homogenization pressure. The lowest T-SH concentration (13.28 ± 0.28 μmol/g) was achieved at 140 MPa. HPH causes a change in the protein structure in which part of the protein molecular structure unfolds, leading to an increasingly disordered structure, and S-SH is converted to -S-S [40]. These denatured proteins can potentially form additional aggregates and form smaller proteins that decrease the T-SH content. The results were in general agreement with the findings of [41,42]. In summary, HPH and the Maillard reaction have a synergistic effect on the sulfhydryl group content.

#### 3.2.3. CD Analysis of Protein Secondary Structure

Far-UV CD measurements are one of the most sensitive techniques for determining the secondary structure of proteins [43]. Protein secondary structures can be divided into four categories: *α*-helices, *β*-sheets, *β*-turns, and random coils. Hydrogen bonds maintain protein secondary structures. Some structural changes cause proteins to become more conformationally flexible, making adsorption at the oil-water interface easier and potentially improving emulsification [44]. Table 1 summarizes the secondary structures of SPI, SPI-GOS-M, and SPI-GOS conjugates. The abundance of *α*-helix and *β*-sheet structures in the SPI-GOS conjugates decreased significantly compared to SPI (*p* < 0.05), reaching a minimum at 140 MPa (α-helix: 15.94 ± 0.02, β-sheet: 20.92 ± 0.01). However, the abundance of *β*-turn and random coil structures increased significantly (*p* < 0.05), reaching a maximum at 0.1 MPa (*β*-turn: 21.35 ± 0.02, random coil: 31.53 ± 0.01). The *α*-helix content of the HPH-treated SPI-GOS conjugates decreased with increasing homogenization pressure. Indeed, the HPH-treated SPI-GOS conjugates had a much lower *α*-helix content than SPI-GOS-0.1. This result suggested that HPH cavitation cleaved protein-peptide chains and covalently bound SPI to GOS, transforming the tight cavity-free structure of the α-helix into a looser structure and thus boosting the molecule’s flexibility [45]. With increasing pressure, the abundance of *β*-sheet and *β*-turn structures decreased, while the random coil structure content increased, implying that HPH impaired the protein’s organized structure and triggered folding [46]. Homogenization can open the peptide chain of SPI molecules, while the number of hydrogen bonds between SPI molecules is reduced following the Maillard reaction, resulting in a decrease in the quantity of *α*-helices and *β*-sheets [47]. The increase in the random coil structure content further indicated that the protein molecular structure unfolded during the Maillard reaction, resulting in a more disordered structure [33]. Taken together, the results indicate that HPH and the Maillard reaction significantly changed the secondary structure of SPI (*p* < 0.05).

#### 3.2.4. FTIR Analysis of Protein Secondary Structure

FTIR spectroscopy is a well-established tool used to establish the secondary structure of proteins [48,49]. The amide I band at 1700–1600 cm^−1^ (C=O stretching) and amide II and III bands at 1550–1500 and 1300–1200 cm^−1^ (C=N stretching and N-H bending), respectively, are indicative of the most important structural properties of proteins [50]. The strength of the absorption peaks differed significantly between SPI, SPI-GOS-M, and SPI-GOS conjugates. As shown in Figure 7, the Maillard reaction consumed some functional groups (NH_2_) and created new compounds (C-H, C=O, C=N) [51]. In contrast to SPI, the amide A band of the SPI-GOS conjugates was red-shifted from 3296 cm^−1^ to 3300 cm^−1^. The spectrum of SPI-GOS-140 displayed the maximum intensity of the amide A band, followed by conjugates treated at 120, 100, 80, and 0.1 MPa. These results suggested that covalent cross-linking was induced in the SPI-GOS conjugates. Meanwhile, the shapes of the hydrogen and amide bonds were changed by HPH and the Maillard reaction [24]. In the spectra of the conjugates, two new peaks were observed at 2883 and 2935 cm^−1^ (C-H stretching), which was compatible with the generation of MRPs [52]. The intensities of the peaks at 1657 and 1541 cm^−1^ in the SPI-GOS conjugate spectrum, which reflected C=O stretching and N-H bending vibrations, were significantly lower than those in the SPI spectrum (*p* < 0.05), suggesting that the Maillard reaction modified the amide II and III bands. Amadori products (C=O) and Schiff bases (C=N) are created when SPI and sugar molecules interact [51]. New peaks emerged at 1313 and 1244 cm^−1^ when the SPI-GOS conjugates were treated with HPH, likely due to covalent bonds forming between the free amino group of SPI and the carbonyl group at the end of the GOS molecule, creating an O-H bending vibration within the sugar circle [20]. Furthermore, the intensities of the absorption peaks at 900 or 1078 cm^−1^ were higher than those observed for SPI, indicating that the GOS molecule was linked to the SPI backbone and further demonstrating the formation of SPI-GOS conjugates [53]. These findings supported the hypothesis that HPH, in combination with the Maillard reaction, significantly altered the structure of SPI (*p* < 0.05).

#### 3.2.5. Intrinsic Fluorescence Analysis of Protein Tertiary Structure

Intrinsic fluorescence spectra based on tryptophan content were used to determine changes in the protein tertiary structure during conjugation [33]. Figure 8 depicts the intrinsic fluorescence spectra of SPI and the SPI-GOS conjugates. The maximum absorption wavelength (λmax) of SPI was 329 nm, which remained unchanged for SPI-GOS-M. Since the sugar molecule has no fluorescence, in this case, it is equivalent to the fluorescence quencher inside the protein molecule, so that the fluorescence intensity is weakened. However, the intrinsic fluorescence was considerably altered by glycation via conventional heating and HPH treatment. The SPI-GOS conjugates showed a red-shift of λmax to varying degrees, with SPI-GOS-0.1 at 331 nm and SPI-GOS-120 at 333 nm. These results demonstrated that tryptophan existed in a polar microenvironment, which enhanced the hydrophilicity of the modified protein [54]. The exposure of hydrophobic groups in proteins and changes in the secondary structure of SPI were thought to be the causes of this occurrence [55]. Furthermore, the fluorescence intensities of the SPI-GOS conjugates were lower than that of SPI because substantial steric hindrance was formed following covalent cross-linking between SPI and GOS to cover the tryptophan chromophores [56]. The combination of HPH and the Maillard reaction disrupted the intermolecular forces in SPI, thus altering the tertiary structure of the protein. At the same time, SPI-GOS-0.1 had a lower fluorescence intensity than SPI-GOS-120, indicating that HPH slowed down the Maillard reaction. Taken together, the results indicate that HPH and the Maillard reaction changed the tertiary structure of SPI.

### 3.3. Effect of HPH on SPI Emulsifying Properties

#### 3.3.1. Emulsifying Activity and Stability

One of the most essential functional features of SPI is its emulsifying properties. Altering SPI by combining HPH with the Maillard reaction improved the protein’s emulsifying capabilities. Figure 9 shows the emulsification activity index (EAI) and emulsion stability index (ESI) of emulsions created with SPI and SPI-GOS conjugates. Both EAI and ESI steadily improved as the homogenization pressure increased, reaching a maximum of 64.63 ± 0.28 m^2^/g and 23.15 ± 0.19 min at 120 MPa, respectively. HPH and the Maillard reaction caused structural unfolding, which yielded greater flexibility and increased spatial site resistance, thus exposing hydrophobic groups contained within the molecule and increasing the lipophilicity of the protein [9]. Therefore, protein molecules may be quickly and tightly adsorbed at the water/oil interface to form a dense protective barrier that prevents oil droplets from aggregating, thus increasing the protein’s emulsifying abilities [57]. Compared with those of SPI, the EAI and ESI values of SPI-GOS-120 were increased by 78.08 and 41.84%, respectively. Notably, when a particular pressure is reached, HPH creates more heat and the mechanical and thermal effects combine to denature the protein, thus reducing EAI and ESI values [9].

#### 3.3.2. Particle Size of Emulsions

The emulsifying properties of a protein can be assessed by the average size and size distribution of droplets in the emulsion [43]. Smaller emulsion droplets generally produce more stable emulsions [18]. Figure 10 and Figure 11 show the average particle size and size distribution of emulsions created with SPI and SPI-GOS conjugates. The average particle size of emulsions created with the conjugates was smaller than that of the emulsion created with SPI. Additionally, the average particle size of emulsions created with HPH-treated SPI-GOS conjugates was smaller than that of the emulsion created with SPI-GOS-0.1. With an increase in homogenization pressure, the average particle size of the emulsions first decreased and then increased. The same tendency was observed for PDI. The minimum average particle size (117 nm) was obtained for the emulsion created with SPI-GOS-120 and the PDI was 0.42. Meanwhile, this treatment generated a single peak in the particle size distribution curve, representing the most stable emulsion. In contrast, the average particle size of the emulsion created with SPI-GOS-140 was 175 nm and several peaks were observed. On the one hand, more MRPs might be adsorbed on the droplet surface, increasing the repulsive force between droplets and decreasing the emulsion particle size [58]. On the other hand, the mechanical stresses associated with HPH can rupture particles, causing a considerable reduction in particle size down to the micron/submicron level [59,60]. Furthermore, excessive pressure can cause tiny protein molecules to. Similarly, both larger aggregates and smaller particles were generated at 100 MPa in a HPH investigation of mussel-isolated proteins [61]. In conclusion, the HPH mechanical force and effect of sugar molecules lead to the change in the structure of the conjugate, which effectively reduces the emulsion particle size and improves the emulsion stability.

#### 3.3.3. Zeta Potential of Emulsions

The zeta potential reflects the surface charge of oil droplets within an emulsion [62,63]. Figure 12 shows the absolute zeta potential values of emulsions created with SPI and SPI-GOS conjugates. The absolute zeta potential values of the SPI-GOS conjugate emulsions were lower than that of the SPI emulsion, indicating that the negative charge on the surface of the SPI-GOS conjugates after the Maillard reaction had a repulsive effect on corresponding particles in the emulsion, thus preventing aggregation between droplets. Greater negative charges create greater intermolecular repulsive forces, which reduce the probability of aggregation between droplets and enhance the emulsion’s stability [64]. The absolute zeta potential values of the emulsions created with the HPH-treated SPI-GOS conjugates tended to increase and then decrease with increasing homogenization pressure. The greatest absolute zeta potential value (23 mV) was obtained for the emulsion created with SPI-GOS-120, indicating that this emulsion was the most stable. Similarly, the emulsion became more stable at 100 MPa in an HPH investigation of a vegetable protein [62]. In contrast, the maximum zeta potential achieved for the emulsion created with SPI-GOS-140 was 18 mV. These results can be attributed to the aggregation of proteins caused by the interaction of the sugar molecules and the mechanical forces of HPH. Taken together, the results indicate that HPH and the Maillard reaction increased the charge of the emulsion droplets, thereby enhancing the stability of emulsions created with SPI-GOS conjugates.

#### 3.3.4. Microscopic Observations of Emulsions

Figure 13 shows representative CLSM images of the emulsions created with SPI and SPI-GOS conjugates. The SPI and SPI-GOS mixture emulsions were composed mainly of relatively large droplets. However, the particle size of emulsions created with the HPH-treated SPI-GOS conjugates decreased and droplet dispersion became more uniform with increasing homogenization pressure. Under the action of high-speed turbulent shear forces and high-pressure differences during HPH, the macromolecules folded to facilitate the combination of SPI and GOS, thus controlling the interface characteristics and reducing the interfacial tension. Moreover, exposure of the hydrophobic groups improved the hydrophilicity and lipophilicity of the proteins, enabling a more even dispersion. In summary, this result is a more inherent demonstration of the effect of the mechanical force of HPH on the Maillard reaction sugar molecules; it can reduce emulsion particle size and evenly distribute the emulsion, thereby improving the EAI and ESI of SPI.

## 4. Conclusions

In this study, SPI was conjugated with GOS via traditional wet heating at 0.1 MPa and under HPH at various homogenization pressures. HPH combined with the Maillard reaction significantly changed the structure and enhanced the emulsifying properties of SPI (*p* < 0.05). HPH slowed the occurrence of the Maillard reaction and controlled the reaction process, as evidenced by lower intermediate MRP content, browning intensity, and free amino acid content. Under HPH at 120 MPa, the secondary structure of the glycated protein was substantially altered in terms of the α-helix and random coil content, as well as amide II vibrational bands. The mechanical forces of HPH dispersed the SPI protein molecules into numerous small molecules. The structural changes improved the functional features of the HPH-treated SPI-GOS conjugates, such as their emulsifying capacity and emulsion stability. In conclusion, HPH and the Maillard reaction during glycation of SPI with GOS can effectively improve the emulsifying properties of SPI. The modified SPI can be used as an emulsifier in the production of plant-based beverages and yogurts, potentially broadening its application in the food industry.

There are two major limitations worth noting in this study. First, due to environmental factors, such as laboratory equipment and site, the basic determination methods used for some indicators have limitations. Second, this study only explored the emulsification, which is only one of the protein functions. In future studies, we will explore the effect of this modification method on other functions of the protein with more in-depth research methods.

## Figures and Tables

**Figure 1 foods-11-03505-f001:**
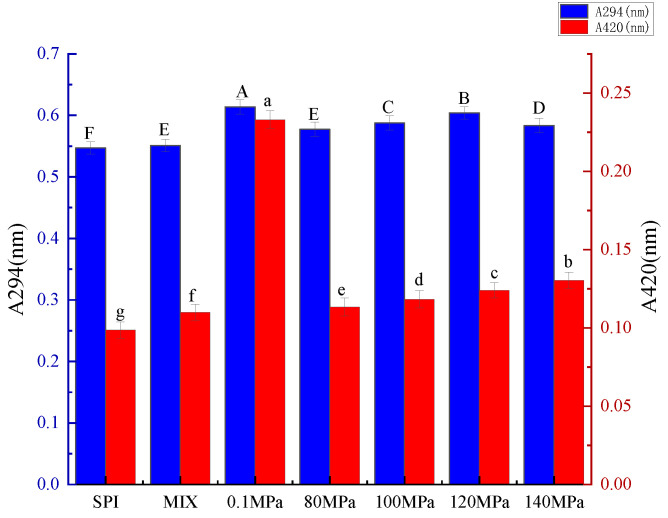
Ultraviolet (UV) absorbance of the samples. A294 nm is indicative of intermediate Maillard reaction products, while A420 nm is indicative of the degree of browning. SPI is the native soy protein isolate, MIX represents the mixture of SPI and GOS, 0.1 MPa represents the traditional wet heating Maillard reaction group, while 80 MPa, 100 MPa, 120 MPa, 140 MPa are the HPH treatment groups. The letters a–g and A–F indicate decreasing orders of magnitude (*p* < 0.05).

**Figure 2 foods-11-03505-f002:**
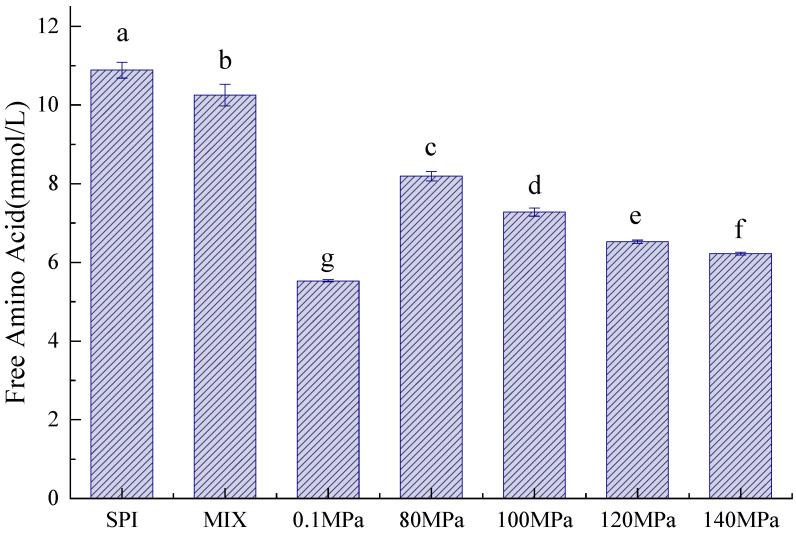
Free amino acid content of the samples. SPI is the native soy protein isolate; MIX represents the mixture of SPI and GOS; 0.1 MPa represents the traditional wet heating Maillard reaction group; 80 MPa, 100 MPa, 120 MPa, and 140 MPa are HPH treatment groups. The letters (a–g) indicate decreasing orders of magnitude (*p* < 0.05).

**Figure 3 foods-11-03505-f003:**
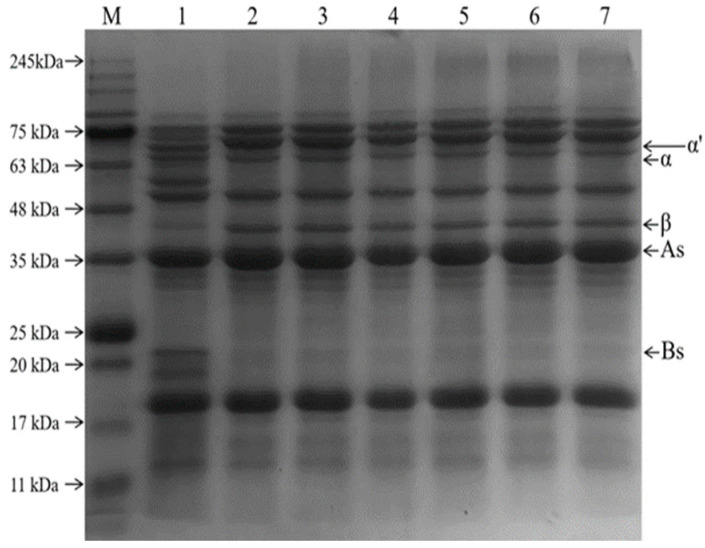
SDS-PAGE patterns of the samples; β-conglycinin (7s): α′, α, and β subunits, glycinin (11s): A and B subunits. Lane M represents molecular weight markers; lane 1 is the SPI; lane 2 represents the mixture of SPI and GOS; lane 3 represents the traditional wet heating Maillard reaction group; lanes 4–7 represent the 80 MPa, 100 MPa, 120 MPa, and 140 MPa HPH treatment groups, respectively.

**Figure 4 foods-11-03505-f004:**
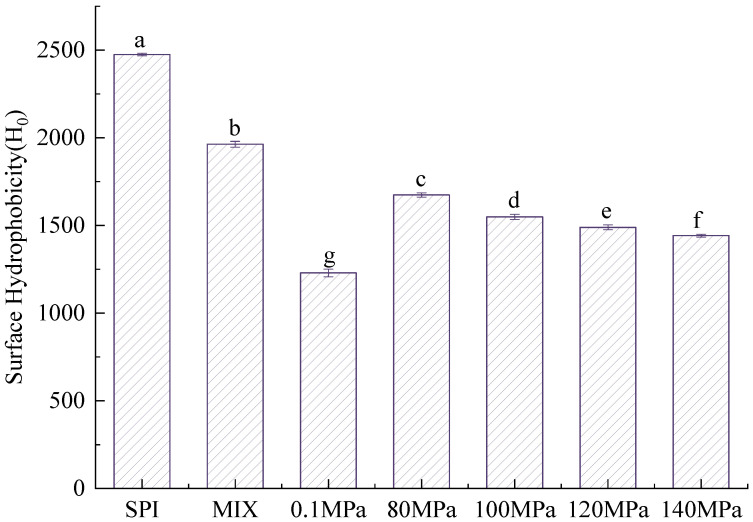
H_0_ values of the samples. SPI is the native soy protein isolate; MIX represents the mixture of SPI and GOS; 0.1 MPa represents the traditional wet heating Maillard reaction group; 80 MPa, 100 MPa, 120 MPa, and 140 MPa are the HPH treatment groups. The letters (a–g) indicate decreasing orders of magnitude (*p* < 0.05).

**Figure 5 foods-11-03505-f005:**
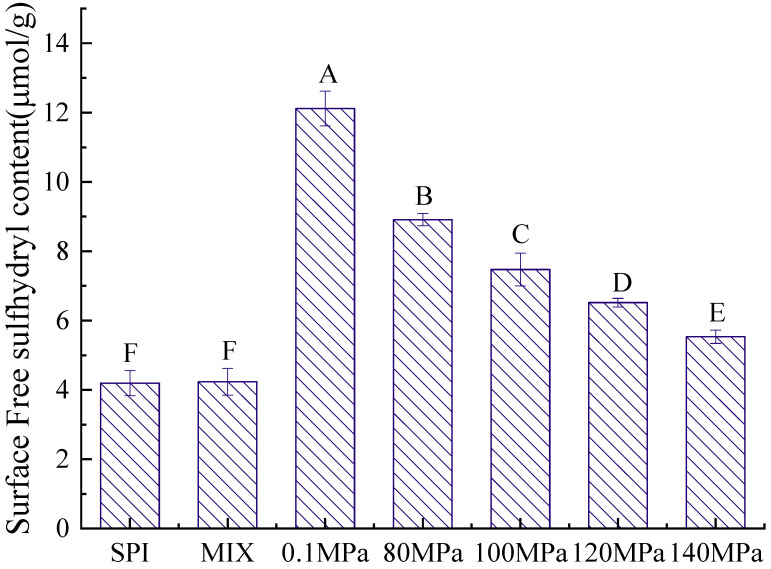
Surface free sulfhydryl content of the samples. SPI is the native soy protein isolate; MIX represents the mixture of SPI and GOS; 0.1 MPa represents the traditional wet heating Maillard reaction group; 80 MPa, 100 MPa, 120 MPa, and 140 MPa are the HPH treatment groups. The letters A–F indicate decreasing orders of magnitude (*p* < 0.05).

**Figure 6 foods-11-03505-f006:**
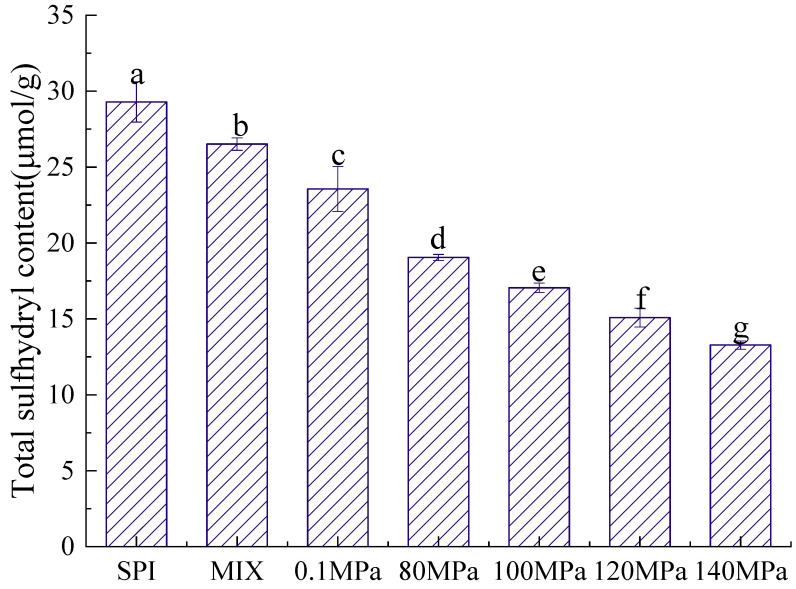
Total sulfhydryl content of the samples. SPI is the native soy protein isolate; MIX represents the mixture of SPI and GOS; 0.1 MPa represents the traditional wet heating Maillard reaction group; 80 MPa, 100 MPa, 120 MPa, and 140 MPa are the HPH treatment groups. The letters (a–g) indicate decreasing orders of magnitude (*p* < 0.05).

**Figure 7 foods-11-03505-f007:**
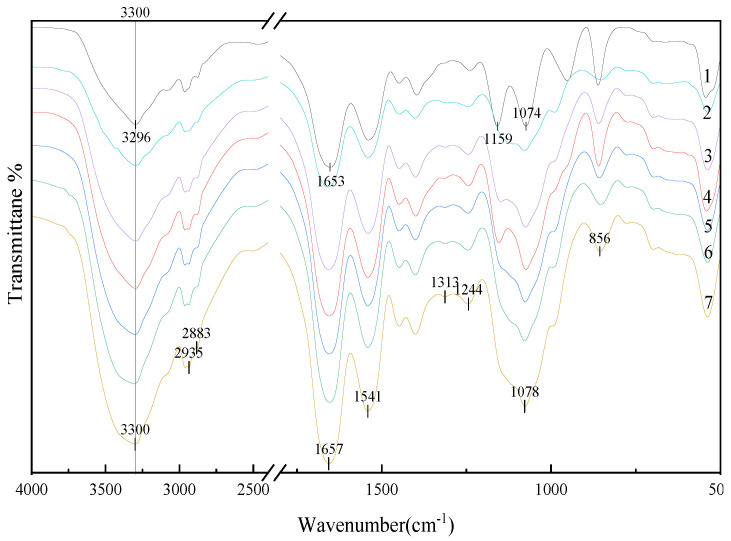
FT-IR spectra of the samples. 1 is the native soy protein isolate; 2 represents the mixture of SPI and GOS; 3 represents the traditional wet heating Maillard reaction group; 4–7 represent the 80 MPa, 100 MPa, 120 MPa, 140 MPa HPH treatment groups, respectively.

**Figure 8 foods-11-03505-f008:**
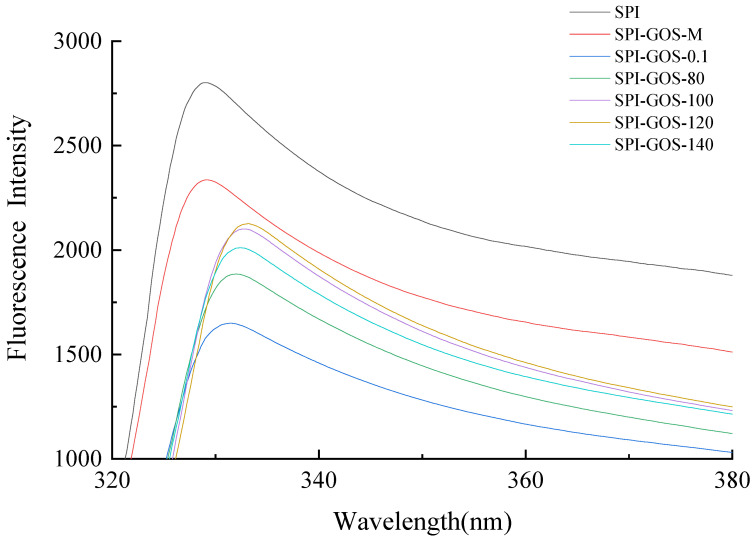
Intrinsic fluorescence spectra of the samples. SPI is the native soy protein isolate; SPI-GOS-M represents the mixture of SPI and GOS; SPI-GOS-0.1 represents the traditional wet heating Maillard reaction group; SPI-GOS-80, SPI-GOS-100, SPI-GOS-120, and SPI-GOS-140 are the HPH treatment groups.

**Figure 9 foods-11-03505-f009:**
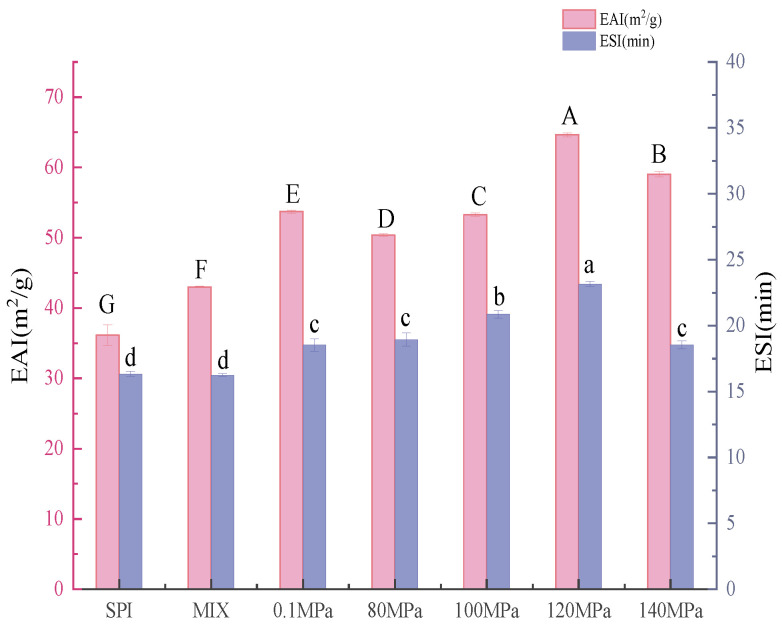
Emulsifying properties of the samples. SPI is the native soy protein isolate; MIX represents the mixture of SPI and GOS; 0.1 MPa represents the traditional wet heating Maillard reaction group; 80 MPa, 100 MPa, 120 MPa, and 140 MPa are the HPH treatment groups. The letters a–d and A–G indicate decreasing orders of magnitude (*p* < 0.05).

**Figure 10 foods-11-03505-f010:**
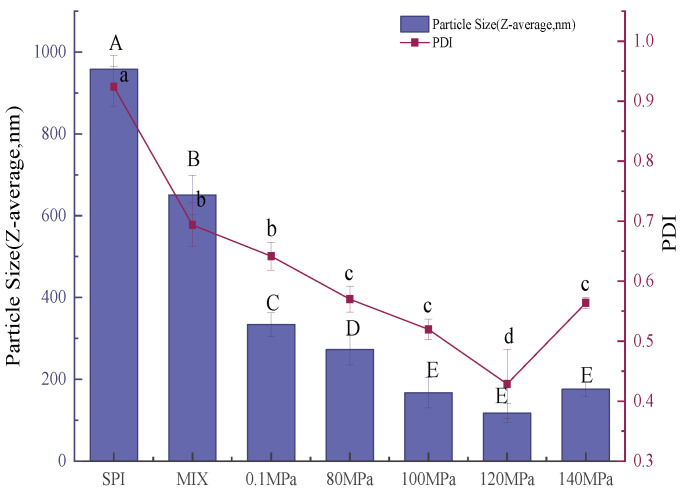
Particle size distribution of emulsions created with the samples. SPI is the native soy protein isolate; MIX represents the mixture of SPI and GOS; 0.1 MPa represents the traditional wet heating Maillard reaction group; 80 MPa, 100 MPa, 120 MPa, and 140 MPa are the HPH treatment groups. The letters a–d and A–E indicate decreasing orders of magnitude (*p* < 0.05).

**Figure 11 foods-11-03505-f011:**
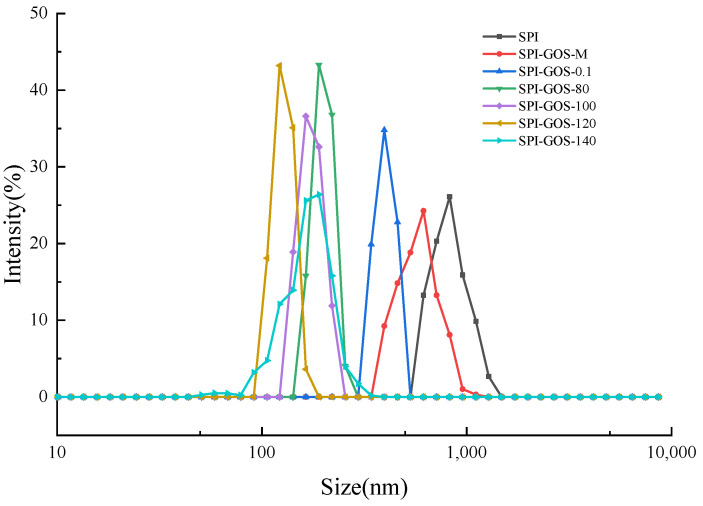
Average particle size of emulsions created with the samples. SPI is the native soy protein isolate; MIX represents the mixture of SPI and GOS; 0.1 MPa represents the traditional wet heating Maillard reaction group; 80 MPa, 100 MPa, 120 MPa, and 140 MPa are the HPH treatment groups.

**Figure 12 foods-11-03505-f012:**
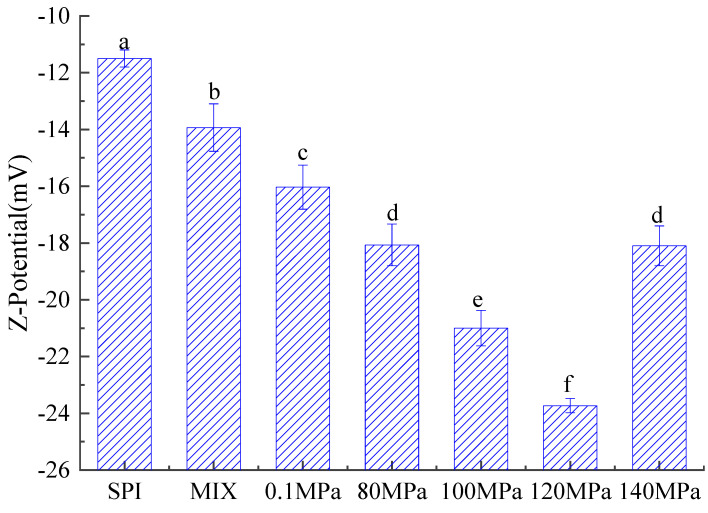
Zeta potential of emulsions created with the samples. SPI is the native soy protein isolate; MIX represents the mixture of SPI and GOS; 0.1 MPa represents the traditional wet heating Maillard reaction group; 80 MPa, 100 MPa, 120 MPa, and 140 MPa are the HPH treatment groups. The letters (a–f) indicate decreasing orders of magnitude (*p* < 0.05).

**Figure 13 foods-11-03505-f013:**
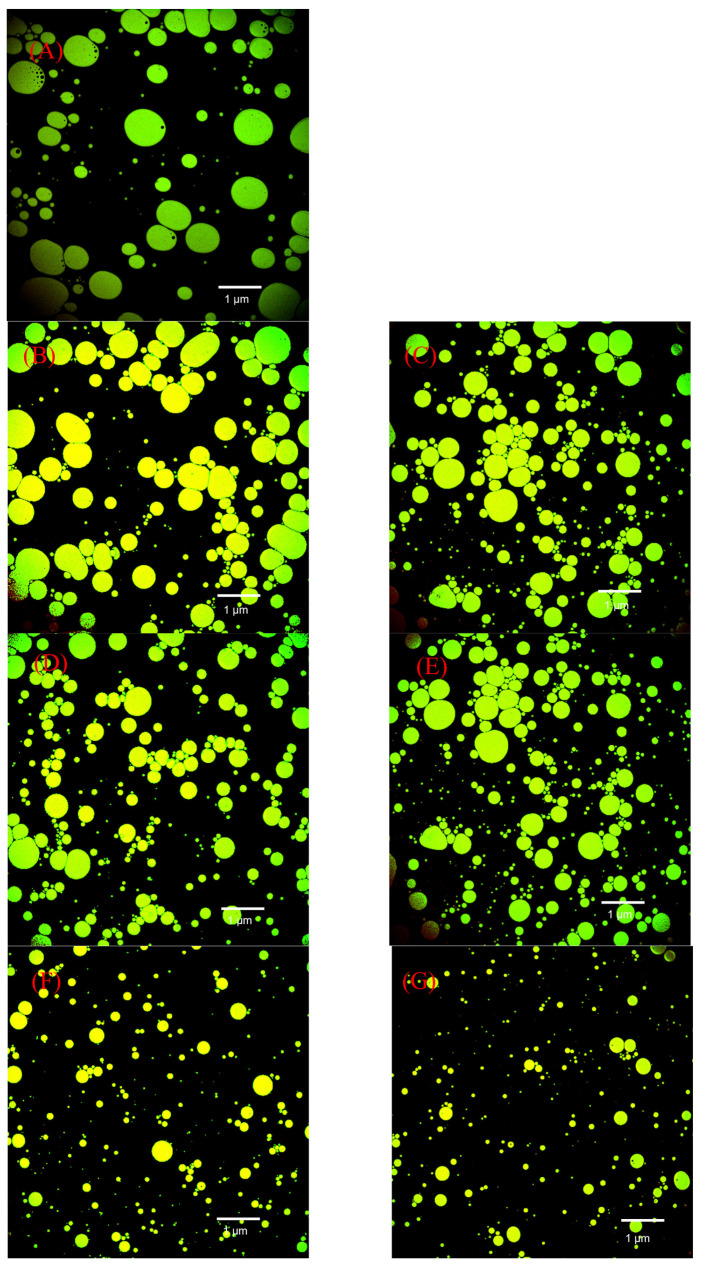
Representative CLSM images of emulsions created with the samples. (**A**) The SPI; (**B**) the mixture of SPI and GOS; (**C**) the traditional wet heating Maillard reaction group; (**D**–**G**) the HPH treatment groups.

**Table 1 foods-11-03505-t001:** Secondary structure composition of glycated SPI formed at different pressures.

Samples	α-Helix (%)	β-Sheet (%)	β-Turn (%)	Random Coil (%)
SPI	33.39 ± 0.02 ^a^	29.34 ± 0.01 ^a^	10.81 ± 0.02 ^f^	26.45 ± 0.01 ^g^
SPI-GOS-M	32.08 ± 0.01 ^b^	28.93 ± 0.01 ^b^	10.65 ± 0.01 ^g^	28.34 ± 0.01 ^f^
SPI-GOS-0.1	19.57 ± 0.02 ^c^	27.55 ± 0.01 ^c^	21.35 ± 0.02 ^a^	31.53 ± 0.01 ^e^
SPI-GOS-80	16.82 ± 0.02 ^d^	24.79 ± 0.01 ^d^	16.26 ± 0.02 ^e^	42.12 ± 0.02 ^d^
SPI-GOS-100	16.15 ± 0.02 ^e^	22.52 ± 0.01 ^e^	17.57 ± 0.01 ^d^	43.76 ± 0.01 ^b^
SPI-GOS-120	16.07 ± 0.01 ^f^	21.53 ± 0.01 ^f^	19.12 ± 0.01 ^c^	43.28 ± 0.01 ^c^
SPI-GOS-140	15.94 ± 0.02 ^g^	20.92 ± 0.01 ^g^	19.32 ± 0.01 ^b^	43.82 ± 0.01 ^a^

SPI is the native soy protein isolate; SPI-GOS-M represents the mixture of SPI and GOS; SPI-GOS-0.1 represents the traditional wet heating Maillard reaction group; SPI-GOS-80, SPI-GOS-100, SPI-GOS-120, and SPI-GOS-140 are the HPH treatment groups. Values represent the mean SD; letters (a–g) indicate significant differences between groups (*p* < 0.05).

## Data Availability

The data that support the findings of this study are available on request from the corresponding author. The data are not publicly available due to privacy or ethical restrictions.

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
