# Peer review of "Structural Characteristics and Emulsifying Properties of Soy Protein Isolate Glycated with Galacto-Oligosaccharides under High-Pressure Homogenization"

_foods, 2022, doi:10.3390/foods11213505_

Round 1

Reviewer 1 Report

This study focuses on determining the structural characteristics and emulsifying properties of soy protein isolate (SPI) glycated with galacto-oligosaccharides (GOS) under high-pressure homogenization. The manuscript is well-written. The experimental design is sound.

Some general comments

1.     Authors should use the common term “soy protein isolate” rather than “soybean protein isolate”.

2.     Emulsification is one of the functional properties of SPI. Why does this study only focus on this functional property? Why are other functional properties not covered? Please justify the selection of this property in this study. 

3.     Section 3.2 discussed physiochemical changes. The title does not reflect this part of the study. I would suggest authors include this in the title. 

4.     All figures and tables are self-explanatory. Any abbreviation, including sample names, should be explained in the caption and/or footnote.

5.     With the improved emulsifying properties, do authors predict that other functional properties of the SPI will also be improved? Authors should state/discuss the potential applications of this study.

6.     Please state the limitations of the study. 

Some specific comments

1.     Line 14: When using the term significant or any other similar meaning, please indicate if this claim is supported by statistical analysis. Use P<0.05 for significantly different and P<0.05 for insignificant differences. If no statistical analysis was carried out, avoid using such terms. Please check the entire manuscript.

2.     Line 15-22: All these changes are significant?

3.     Line 28: Purest form? I would suggest using a different term for this sentence. The word pure means not mixed with other substances.

4.     Line 39-40: Do you mean melanoidins?

5.     Line 46 & 62: Please check the format for the citations.

6.     Line 81, 106, 119, 126, 149, 163, 174: What modifications were made to the preparation method?

7.     Line 87: Why use Biuret Method and not the AOAC method?

8.     Line 94-96: What is the role of the traditional wet heating method? Is this heating condition sufficient to trigger Maillard reaction? Kindly provide references for the method.

9.     Line 179: What are T and n representing?

10.  Fig. 1: Avoid using a line graph for these results since the data are not continuous, at least for SPI, MIX, and 0.1MPa. Please provide the standard deviations.

11.  Line 265: This sentence is a bit misleading. Some of the bands in these regions faded, but some remained unchanged. Please explain more specifically the changes on these bands that can be observed in Fig. 3.

12.  Line 334: Please reconfirm if it is (p<0.5). Please check the entire manuscript.

Author Response

Itemized list of changes addressing reviewer comments

Reviewer #1:

This study focuses on determining the structural characteristics and emulsifying properties of soy protein isolate (SPI) glycated with galacto-oligosaccharides (GOS) under high-pressure homogenization. The manuscript is well-written. The experimental design is sound.

Some general comments

  1. Authors should use the common term “soy protein isolate” rather than “soybean protein isolate”.

Response: Thank you for your comments.

According to your comments, “soybean protein isolate” has been replaced by “soy protein isolate” in the manuscript.

  1. Emulsification is one of the functional properties of SPI. Why does this study only focus on this functional property? Why are other functional properties not covered? Please justify the selection of this property in this study.

Response: Thank you for your comments.

In the preliminary experiment, we found that the functional properties of SPI, including solubility and emulsifying properties, were well improved by this modification method, and emulsifying properties were the best. Therefore, we carried out in-depth study on this property.

  1. Section 3.2 discussed physiochemical changes. The title does not reflect this part of the study. I would suggest authors include this in the title.

Response: Thank you for your comments.

According to your comments, the title of section 3.2 has been changed to “Effect of HPH on changes SPI-GOS conjugate physicochemical properties”.

  1. All figures and tables are self-explanatory. Any abbreviation, including sample names, should be explained in the caption and/or footnote.

Response: Thank you for your comments.

According to your comments, all figures and tables are modified as required.

  1. With the improved emulsifying properties, do authors predict that other functional properties of the SPI will also be improved? Authors should state/discuss the potential applications of this study.

Response: Thank you for your comments.

In our previous experiments, we found that the solubility of SPI was increased, so we predicted that the foaming property would also be improved. The modified SPI can be used as an emulsifier in the production of plant-based beverages and yogurts. This discussion has been modified in the conclusion. The specific contents are as follows.

In conclusion, HPH and the Maillard reaction during glycation of SPI with GOS can effectively improve the emulsifying properties of SPI. The modified SPI can be used as an emulsifier in the production of plant-based beverages and yogurts, potentially broadening its application in the food industry.

  1. Please state the limitations of the study.

Response: Thank you for your comments.

There are two major limitations worth noting in this study. First, due to environ-mental factors, such as laboratory equipment and site, the basic determination methods used for some indicators have limitations. Second, this study only explored the emulsification, which is only one of the protein functions. In future studies, we will explore the effect of this modification method on other functions of the protein with more in-depth research methods.

Some specific comments

  1. Line 14: When using the term significant or any other similar meaning, please indicate if this claim is supported by statistical analysis. Use P<0.05 for significantly different and P<0.05 for insignificant differences. If no statistical analysis was carried out, avoid using such terms. Please check the entire manuscript.

Response: Thank you for your comments.

According to your comments, I have checked the entire manuscript, and revised them accordingly.

  1. Line 15-22: All these changes are significant?

Response: Thank you for your comments.

Yes, all these changes are significant. I have revised them accordingly.

  1. Line 28: Purest form? I would suggest using a different term for this sentence. The word pure means not mixed with other substances.

Response: Thank you for your comments.

The sentence has been revised to “Soy protein isolate (SPI) is a commercially important biopolymer with a protein concentration ≥ 90% on a dry weight basis”.

  1. Line 39-40: Do you mean melanoidins?

Response: Thank you for your comments.

Yes, it means melanoidins.

  1. Line 46 & 62: Please check the format for the citations.

Response: Thank you for your comments.

According to your comments, I have checked the format for the citations, and revised them accordingly.

  1. Line 81, 106, 119, 126, 149, 163, 174: What modifications were made to the preparation method?

Response: Thank you for your comments.

According to the samples’ properties and the experimental equipment’s state, the sample concentration and detection dosage were adjusted accordingly.

  1. Line 87: Why use Biuret Method and not the AOAC method?

Response: Thank you for your comments.

Biuret Method is a commonly used for protein content determination in laboratories, and the data obtained are fast and accurate, so the Biuret Method was selected.

  1. Line 94-96: What is the role of the traditional wet heating method? Is this heating condition sufficient to trigger Maillard reaction? Kindly provide references for the method.

Response: Thank you for your comments.

This experimental group was set up to compare the state of the atmospheric Maillard reaction under the same high-pressure homogeneous reaction condition, and it has been shown in the electrophoretic diagram that the Maillard reaction occurred under this condition.

  1. Line 179: What are T and n representing?

Response: Thank you for your comments.

According to your comments, this part has been revised. The specific contents are as follows.

where T is constant with a value of 2.303; n is the dilution ratio of 100; c is the protein con-centration before emulsion formation (g/mL); and U is the oil volume fraction 0.25.

  1. Fig. 1: Avoid using a line graph for these results since the data are not continuous, at least for SPI, MIX, and 0.1MPa. Please provide the standard deviations.

Response: Thank you for your comments.

According to your comments, Fig.1 has been revised. The specific contents are as follows.

  1. Line 265: This sentence is a bit misleading. Some of the bands in these regions faded, but some remained unchanged. Please explain more specifically the changes on these bands that can be observed in Fig. 3.

Response: Thank you for your comments.

According to your comments, I have checked the entire manuscript, and revised them accordingly. The specific contents are as follows.

When the pressure was 0.1-140 MPa, a new band at the upper end of the gel was observed, and the band deepened with the increase in pressure. It indicated that SPI and GOS underwent the Maillard reaction to create the large molecular weights of the conjugates, which are too large to migrate into the separation gel. In addition, the dis-appearance of the 60 kDa and 19 kDa bands indicated that the subunits of the corresponding molecular weight proteins covalently combined with the sugar molecules to form the conjugate.

  1. Line 334: Please reconfirm if it is (p<0.5). Please check the entire manuscript.

Response: Thank you for your comments.

According to your comments, I have checked the entire manuscript, I have checked the entire manuscript, and revised them accordingly.

Reviewer 2 Report

The paper is of good scientific quality. The effect of glycation has been investigated with different techniques. I suggest some minor modifications:

-pag.2 line 89: even if the biuret is an old method, I think the reference is necessary.

-pag.7 line 272. Figure 3 legend. Line 1, cited in the text, is not present in the legend.

- A comment on the SDS-PAGE figure 3: the lines 1 and 2 pattern comparison shows some differences, SPI-GOS-M was not treated, the mixing between SPI and GOS caused minor differences, and the band at 60 kDa and 19 kDa disappear. I think it is necessary to comment. The small modifications are supported by the change in hydrophobicity (fig.4), by total sulfhydryl content (Fig.6), by structure (table 1), and by all the other experiments (Fig. 7-8-9-10-11-12-13). In fig. 8 in SPI-GOS-M there is not a red shift, but a decrease in fluorescence intensity. In fig. 11-12-13, the molecular properties of the droplet are different. The mechanical force and the presence of GOS determined a little structural modification.

-pag.10 line 378: ‘absorption peaks at 857 or 1064 cm-1 were higher than...’ around 900 and not 856, from the figure, there is a pick shift in sample 2. The pick 1078 increased.

-pag.10 fig. 7: the figure ordinate scale is A or T % is not reported.

Author Response

Reviewer #2:

The paper is of good scientific quality. The effect of glycation has been investigated with different techniques. I suggest some minor modifications:

-pag.2 line 89: even if the biuret is an old method, I think the reference is necessary.

Response: Thank you for your comments.

The reference has been added.

-pag.7 line 272. Figure 3 legend. Line 1, cited in the text, is not present in the legend.

Response: Thank you for your comments.

According to your comments, Figure 3 has been revised. The specific contents are as follows.

Lane M represents molecular weight markers; lane 1 is the SPI; lane 2 represents the mixture of SPI and GOS; lane 3 represents the traditional wet heating Maillard reaction group; lanes 4, 5, 6, and 7 represent the 80 MPa, 100 MPa, 120 MPa, and 140 MPa HPH treatment groups, respectively.

- A comment on the SDS-PAGE figure 3: the lines 1 and 2 pattern comparison shows some differences, SPI-GOS-M was not treated, the mixing between SPI and GOS caused minor differences, and the band at 60 kDa and 19 kDa disappear. I think it is necessary to comment. The small modifications are supported by the change in hydrophobicity (fig.4), by total sulfhydryl content (Fig.6), by structure (table 1), and by all the other experiments (Fig. 7-8-9-10-11-12-13). In fig. 8 in SPI-GOS-M there is not a red shift, but a decrease in fluorescence intensity. In fig. 11-12-13, the molecular properties of the droplet are different. The mechanical force and the presence of GOS determined a little structural modification.

Response: Thank you for your comments.

(1)Section 3.1.3 has been revised. The specific contents are as follows.

When the pressure was 0.1-140 MPa, a new band at the upper end of the gel was observed, and the band deepened with the increase in pressure. It indicated that SPI and GOS underwent the Maillard reaction to create the large molecular weights of the conjugates, which are too large to migrate into the separation gel. In addition, the disappearance of the 60 kDa and 19 kDa bands indicated that the subunits of the corresponding molecular weight proteins covalently combined with the sugar molecules to form the conjugate.

(2)Fig. 4, Fig. 6, and all the other experiments (Fig. 7-8-9-10-11-12-13) have been revised.

(3)Figure 8 have been explained in the manuscript. The specific contents are as follows. 

Since the sugar molecule has no fluorescence, in this case, it is equivalent to the fluorescence quencher inside the protein molecule, so that the fluorescence intensity is weakened. It has been explained in the manuscript.

(4)Figure 11 have been explained in the manuscript. The specific contents are as follows.

In conclusion, the HPH mechanical force and effect of sugar molecules lead to the change in the structure of the conjugate, which effectively reduces the emulsion parti-cle size and improves the emulsion stability.

Figure 12 have been explained in the manuscript. The specific contents are as follows. 

These results can be attributed to the aggregation of proteins caused by the interaction of the sugar molecules and the mechanical forces of HPH.

Figure 13 have been explained in the manuscript. The specific contents are as follows.

In summary, this result is a more inherent demonstration of the effect of the mechanical force of HPH on the Maillard reaction sugar molecules, can reduce emulsion particle size and evenly distribute the emulsion, thereby improving the EAI and ESI of SPI.

-pag.10 line 378: ‘absorption peaks at 857 or 1064 cm-1 were higher than...’ around 900 and not 856, from the figure, there is a pick shift in sample 2. The pick 1078 increased.

Response: Thank you for your comments.

According to your comments, this part has been revised. The specific contents are as follows.

Furthermore, the intensities of the absorption peaks at 900 or 1078 cm-1 were higher than those observed for SPI, indicating that the GOS molecule was linked to the SPI backbone and further demonstrating the formation of SPI-GOS conjugates.

-pag.10 fig. 7: the figure ordinate scale is A or T % is not reported.

Response: Thank you for your comments.

According to your comments, this part has been revised. The specific contents are as follows.
